# Nutrition and Rheumatoid Arthritis Onset: A Prospective Analysis Using the UK Biobank

**DOI:** 10.3390/nu14081554

**Published:** 2022-04-08

**Authors:** Camilla Barbero Mazzucca, Lorenza Scotti, Giuseppe Cappellano, Francesco Barone-Adesi, Annalisa Chiocchetti

**Affiliations:** 1Dipartimento di Scienze Della Salute, Interdisciplinary Research Center of Autoimmune Diseases—IRCAD, Università del Piemonte Orientale, 28100 Novara, Italy; camilla.barbero@uniupo.it (C.B.M.); giuseppe.cappellano@med.uniupo.it (G.C.); 2Center for Translational Research on Autoimmune and Allergic Diseases—CAAD, Università del Piemonte Orientale, 28100 Novara, Italy; 3Dipartimento di Medicina Traslazionale, Università del Piemonte Orientale, 28100 Novara, Italy; lorenza.scotti@uniupo.it (L.S.); francesco.baroneadesi@uniupo.it (F.B.-A.)

**Keywords:** dietary habits, autoimmunity, rheumatoid arthritis, UK biobank

## Abstract

Rheumatoid arthritis (RA) is a chronic inflammatory disease that affects the joints. The multifactorial etiopathogenesis of RA has been heavily investigated, but is still only partially understood. Diet can represent both a risk factor and a protective factor, based on some evidence that suggests specific properties of certain foods and their ability to increase/reduce inflammation. To date, the studies done on this topic provide discordant results and are heterogeneous in terms of design and cohort size. In this work, we investigated for the first time the relationship between nutrition and the risk of RA onset using a sample size of about half a million subjects from one of the largest publicly available biobanks that is the UK biobank. Results showed that oily fish, alcohol, coffee and breakfast cereals have protective roles in RA; whereas, tea can increase the risk of RA. In conclusion, the obtained results confirm that diet plays key roles in RA, either by promoting or by preventing RA onset and development. Future research should focus on unravelling the effects of dietary habits on immune-mediated diseases to establish better preventive strategies.

## 1. Introduction

Rheumatoid arthritis (RA) is a common systemic inflammatory disorder characterized by synovial inflammation and swelling. Given the presence of autoantibodies, such as rheumatoid factor (RF) and anti–citrullinated protein antibody (ACPA), RA is considered an auto-immune disease (AD) [1], and it remains the most frequently diagnosed systemic inflammatory arthritis. Its prevalence varies between 0.3% and 1%, mainly occurring during the most productive years of adulthood, that is, between the ages of 20 and 40 [2]. Being characterized by synovial hyperplasia, joint tenderness and joint destruction, it has significant impacts on patients’ physical, emotional and social functioning. It can also culminate into severe disability and premature mortality. Indeed, RA patients present a higher risk of cardiovascular and other systemic complications [3]. Though different phenotypes of RA have been shown, the therapeutic approach mainly relies on a “trial and error” pattern, since no prognostic biomarkers for response to targeted therapies exist, and late diagnosis decreases the chances of remission. For this reason, the identification of early biomarkers and modifiable factors is crucial [4]. RA is considered a multifactorial disease, and it is difficult to exactly determine the respective contribution of genes and environmental factors in its etiology. The role of genetic predisposition is highlighted by the higher incidence of RA in patients with a family history of the disease [5]. Among environmental determinants, smoking and other airborne exposures, microbiota, infectious agents and socioeconomic conditions have been identified as potential predisposing factors [4]. As suggested by different studies in nutritional immunology, dietary habits may either reduce or increase the susceptibility to RA, given the powerful effect of different combinations of nutrients in regulating the immune system [6].

The dietary pattern known as “Western Diet” (WD) is characterized by a high intake of red meat, ultra-processed foods, refined carbohydrates and a low ratio of omega-3 to omega-6 fatty acids. According to recent studies, WD might increase the risk of RA in two ways: on one side by directly boosting inflammation, and on the other by indirectly increasing insulin resistance and obesity [7]. This would partially explain why the prevalence of other ADs, in general, and RA in particular, seems to be higher in the industrialized countries [8]. 

Within the overall WD, specific foods have been associated with RA risk. This is the case of sugar-sweetened soda consumed on a regular basis [9], or beverages containing an excess of free fructose [10], which increase the risk to develop seropositive and all-type RA, respectively. Moreover, sodium, which is heavily present in ultra-processed foods, has been shown to exacerbate the effect of smoking as a risk factor for ACPA-positive RA patients [11].

Besides high sugar and salt, WD is also characterized by an excessive consumption of animal source food. Meat, dairy, and eggs contain choline and carnitine that are then metabolized by the gut microbiota to trimethylamine N-oxide (TMAO) [12]. This compound is in turn associated with an increase in systemic inflammation, and with the risk of atherosclerosis [13]. It is noteworthy that TMAO is naturally present in fish and crustaceans. Meat, eggs and dairy products represent a source of arachidonic acid (AA) and omega-6 fatty acids, which mainly have a pro-inflammatory role, being precursors of lipid mediators of inflammation, namely prostaglandins and leukotrienes, which are, together with neutrophils, the big mediators of the acute inflammatory response [14]. Several studies have been carried out to assess whether the consumption of meat might contribute to the risk of developing RA. Unfortunately, the results are inconsistent since some studies highlight the harmful effect of meat consumption [15], while others highlight the absence of any significant association [16,17]. 

Similarly, the effect of dairy consumption on the risk of developing RA is unclear. A protective effect, mainly related to its vitamin D content, has been suggested [18]; however, another study found no significant associations between dairy product consumption and RA risk [15]. It is noteworthy that this micronutrient’s deficiency is associated with another autoimmune condition, namely multiple sclerosis (MS) [19]. 

Despite the potential protective role of fish with respect to RA onset that is mediated by the anti-inflammatory properties of n-3 polyunsaturated fatty acids (PUFA), there is evidence that disprove its beneficial effect [20], due to the positive correlation observed between fish consumption and urinary TMAO levels [21]. Accordingly, the dose–response approach adopted by the authors of a large meta-analysis could be more appropriate than a mere comparison of extreme categories of consumption (high versus low) [22]. This opens the possibility to define an appropriate and healthy range of foods intake, rather than a univocal role for individual foods. 

A moderate fish consumption and an adequate intake of omega-3 (also derived from nuts and seeds) are hallmarks of the Mediterranean Diet (MD), which is also characterized by a frequent consumption of olive oil, whole grains, and seasonally available fruit and vegetables, as well as a moderate intake of dairy, meat, many condiments and spices, water, infusions and wine. The MD has been indicated for the prevention of numerous chronic diseases, given its anti-inflammatory properties [23], which derive not only from the more balanced omega-6 to omega-3 ratio, but also from the high intake of polyphenols, fiber and vitamins contained in the foods typical of this diet [6]. A recent meta-analysis showed that the MD, together with other anti-inflammatory dietary patterns, is more effective in reducing pain and increasing physical function in RA compared to ordinary diets; moreover, the MD demonstrated to have a greater efficacy in reducing pain in comparison with vegetarian and vegan diets following subgroup analysis [24]. However, a positive effect of the MD in reducing the risk of RA onset could not, so far, be demonstrated [5,6]. A recent large, population-based prospective cohort study of French women found an inverse association between adherence to the MD and RA risk among ever-smokers, but not among never-smokers [25]. In another population-based case control study, an association between high MD score and low risk of RA was observed in RF-positive and in ACPA-positive RA patients [26]. 

Many studies have been conducted specifically for single foods highly represented in this dietary pattern, such as vegetables, alcohol, coffee, and tea. The role of vegetable consumption remains unclear, with evidence suggesting an inverse association with RA risk [27] and the absence of significant correlations [28]. Additionally, the impact of coffee and tea consumption on RA risk is not well-defined. Regarding tea, there are studies indicating, respectively, a null [15,29] and a protective effect with respect to disease risk [30]. For coffee, both the absence of correlation [29] and the positive association with RA [31], seropositive RA [32] and rheumatoid factor (RF) [33] has been suggested. From the state-of-the-art analysis, alcohol remains one of the few food items whose protective effect has been confirmed with more certainty [34,35]. This underlines the urgency to further explore the relationship between diet and RA risk in larger cohorts. 

In the present work, we analyzed the effect of diet on RA onset exploiting one of the largest databases available to date, that is, the United Kingdom (UK) biobank. This is a publicly available biobank study conducted in the UK, which is investigating the role of genetic predisposition and exposure to environmental factors (including dietary habits and lifestyle) in the development of different human diseases. The UK biobank collects data from 500,000 volunteers in the UK whose ages range between 40 and 69 years; each participant was interviewed about lifestyle, medical history and nutritional habits and provided blood and urine samples [36]. 

## 2. Methods

### 2.1. Cohort Selection

This is a prospective cohort study based on the data collected from the UK Biobank. The initial sample consisted of 502,507 subjects aged 40–69 years, living in England, Wales or Scotland, who accepted to participate in the UK biobank cohort study in 2006 (pilot phase) and 2007–2010 (main phase). Subjects were identified from the National Health Service (NHS) patient register and were invited by email to attend the first assessment visit at a center located within 25 km from their residence. During the first assessment visit, subjects were asked to complete a touchscreen questionnaire aimed at collecting information regarding sociodemographic characteristics, lifestyle habits, medical history and dietary habits, among others. For the main study, subjects were excluded based on the following exclusion criteria: prevalent cases (affected by RA before cohort entry), subjects who withdraw their consent to participate in the study or had missing information on all dietary factors, and subjects who self-reported RA anytime during the study period, since the diagnosis date was unknown and only the estimated year of RA onset was available. The presence of RA was identified as a hospital admission with primary or secondary diagnosis of RA (International Classification of Disease (ICD)-9 code 714.* or ICD-10 codes M05 and M06), while 6605 subjects self-reported diagnosis of RA (code 1464).

Associations will be further explored comparing the distribution of food frequency levels of incident and prevalent cases in a secondary analysis conducted a posteriori (Figure 1).

### 2.2. Dietary Data Collection

Starting from 2006 (baseline) until 2010 (end of the enrollment phase) dietary habits of study participants were assessed through a food frequency questionnaire (FFQ), reporting their usual intake over the last year of 29 different food groups and alcohol. Specifically, participants were asked about their daily intake of cooked vegetables, salad/raw vegetables, fresh fruit, dried fruit, coffee and tea, as well as their weekly consumption of oily fish, other fish, processed meats, poultry, beef, lamb, pork, cheese, salt added to food, tea, water, milk, type of spread, bread and breakfast cereals [37]. The type of coffee consumed (decaffeinated, ground, instant and other) was also assessed. For daily intake, subjects could report the exact number of tablespoons/serving consumed daily, whereas for weekly intake they could choose one of the following options: never, less than once a week, once a week, 2–4 times a week, 5–6 times a week or once or more daily. When possible, the frequency of intake of each food group was collapsed to obtain at least three classes, each including at least 10% of the total number of subjects. This allowed the evaluation of a trend in RA risk related to elevated levels of consumption.

Starting from 2009, the main study protocol was enhanced with the Oxford WebQ, a web-based 24 h dietary assessment tool asking about the consumption of up to 206 types of foods and 32 types of drinks during the previous 24 h. This interview was aimed at estimating the actual dietary habits and was completed at the assessment center by participants enrolled between April 2009 and September 2010. After the recruitment period, an email was also sent out every 3–4 months, between February 2011 and June 2012 asking participants to autonomously repeat the 24 h recall from home. 

In the present study, we only considered data derived from FFQ since our aim was to investigate the usual rather than the actual dietary habits of study participants. The latter may represent a limitation of our study since data from the 24 h interview available for a subsample of participants could be a valuable resource to deepen the preliminary data deriving from the present study. However, the long-term reproducibility of the main dietary touchscreen variables, as well as the capability of the FFQ to reliably rank participants according to the intake of main foods, was confirmed by researchers of the Nuffield Department of Population Health, University of Oxford [37].

### 2.3. Outcome Ascertainment

Information on subjects’ hospitalization were retrieved through record linkage with NHS hospital in-patient data from hospital episode statistics in England, the Scottish Morbidity Records, and the Patient Episode Database for Wales. Subjects hospitalized with primary or secondary diagnosis of RA identified by ICD-10 codes M05 and M06 were classified as events.

Subjects were followed from the date of the first assessment visit, during which nutritional information was collected, until the earliest date on which one of the following events take place: (i) hospitalization with diagnosis of RA, (ii) death, (iii) migration and (iv) end of follow-up (31 March 2017).

### 2.4. Covariates Assessment

The information obtained at the first assessment visit were selected as potential confounders of the relationship between diet and risk of RA, and these are: age (<50, [50–60), [60–70), ≥70) sex, study area (England, Scotland, Wales), Townsend deprivation index (quartiles: ≤−3.65, (−3.65; −2.16], (−2.16; 0.49], >0.49), smoking (current smokers, current occasional smokers, past smokers, past occasional smokers, never smokers), body mass index (BMI, Underweight (<18.5), normal weight (18.5–24.9), overweight (25.0–29.9) obese (≥30.0)), number of days/week performing moderate or vigorous physical activity (0, 1–3, 4–6, 7 days/week), comorbidities (type II diabetes, hypertension, hypercholesterolemia and other autoimmune diseases, namely, psoriasis, multiple sclerosis, systemic lupus erythematosus and inflammatory bowel disease), and alcohol consumption (≤3 times a month, 1–4 times week, daily or almost daily). Townsend deprivation index measures the degree of socioeconomic deprivation based on data regarding car ownership, household overcrowding, owner occupation, and unemployment. The information regarding these aspects were derived from the national census and aggregated by postcodes of residence. Higher values of the index are suggestive of higher socioeconomic deprivation [38]. Alcohol consumption was considered both as a confounder and as a diet component in the different analyses.

### 2.5. Statistical Analysis

Descriptive statistics were performed to summarize the main characteristics of the subjects included in the study and dietary habits. Categorical variables were reported as absolute frequencies and percentages. Univariable and multivariable Cox proportional hazard models were fitted to estimate the hazard ratios (HR) and the corresponding 95% confidence intervals (95%CI) for the association between dietary factors and risk of RA. The lowest consumption level was considered as a reference category. Multivariable models were adjusted for age, sex, study area, Townsend deprivation index, smoking, BMI, moderate/vigorous physical activity and alcohol consumption. The test for trend was also performed for the multivariable model. Logistic regression was used to calculate the prevalence odds ratios (pOR) and the corresponding 95%CI to compare the prevalence of the intake of selected food groups between incident and prevalent RA cases and prevalent RA cases.

## 3. Results

From the 502,507 initial subjects included in the UK biobank, 18 were excluded because they withdrew their consent to participate in the study, while 6605 subjects were excluded because they were already affected by RA. Furthermore, 16,259 and 131 subjects were excluded because of missing data on diet and self-reported diagnosis of RA during follow-up, respectively. Thus, the final cohort consisted of 479,494 subjects. Cohort members accumulated 3,905,828 year of follow-up (on average 8.15 year per subject) and generated 2819 events of RA, leading to an incidence rate of 7.22 RA events per 10,000-person year. 

Figure 2 shows the flow-chart of the exclusion criteria.

Table 1 shows the overall distribution of the cohort by main characteristics, as well as by event status, the HR and corresponding 95%CI obtained from the univariable Cox regression model. All variables considered are associated with RA onset. As expected, male RA patients had a lower risk of developing RA (HR 0.587, 95%CI 0.543–0.635). Decreasing levels of Townsend deprivation index (where low levels represent a high socioeconomic status) and increasing levels of physical activity were associated to a decrease in the risk of RA, with HR ranging from 0.754 for the third quartile of Townsend deprivation index to 0.623 for the first quartile compared to the fourth quartile and from 0.507 for 1–3 day a week of physical activity to 0.732 for 7 days a week compared to 0 days/week. Weekly or daily alcohol consumption were associated with a 40% decreased risk of RA compared to subjects drinking less than three times per month. Increasing age and increasing levels of BMI were associated with an increased risk of RA, which is similar to elevated frequency of smoking (current smoker vs. never smokers HR 1.779, 95%CI 1.569–2.016). The distribution of missing values for all variables is reported in the Appendix A. Table 2 describes the distribution of the frequency of consumption of different food groups, the unadjusted and adjusted HR, and the corresponding 95%CI for the association between food intake and risk of RA. 

The food group of dried fruit was not included in the analyses because of the few consumers among RA events. High consumption of tea (≥4 cups/day) was associated with an increased risk of RA 23%, compared to less than 1 cup/day. We also found a statistically significant trend in regard to tea intake (*p*-value < 0.0001). Conversely, oily fish intake, compared to no consumption, was associated with a slight decrease in the risk of RA with adjusted HR (aHR) of 0.803 (95%CI 0.707–0.913) for less than once-a-week consumption, 0.782 (95%CI 0.689–0.887) for once-a-week intake and 0.862 (95%CI 0.750–0.990) for a consumption of at least two times a week, though no statistically significant differences were found. High consumption of cheese (≥4 times/week) and breakfast cereals (≥6 times/week) were associated to a decreased risk of RA compared to low consumption, with aHR equal to 0.802 (95%CI 0.694–0.926) and 0.866 (95%CI 0.781–0.959), respectively. For both food groups, a statistically significant trend was found, suggesting that RA risk decreases for increasing levels of consumption. 

Weekly or daily alcohol consumption was associated with a reduced risk of RA compared to the consumption less than 3 times per month, respectively of 24% and 26%. A protective effect of coffee consumption on RA risk was observed for subjects drinking 1–2 cups per day compared to sporadic or nondrinkers (aHR 0.838, 95%CI 0.763–0.920); however, no statistically significant reduced risk was observed for high intake. Finally, although no statistically significant increased risk was observed, an unexpected significant trend was observed for cooked vegetables suggesting a potential increased risk of RA for increasing frequency of intake. Results for food groups associate to RA onset are also reported in Figure 3. 

To reinforce the above associations, we compared the prevalence of intake levels of cheese, breakfast cereals, oily fish and tea of incident and prevalent cases (Table 3); we did not find any significant difference in the consumption of the considered food groups among those subjects who were already affected by RA before FFQ completion and those who developed RA after its completion. 

## 4. Discussion

Lifestyle factors and diet have been previously indicated as environmental RA risk modifiers [39]. Our results, indicating a positive correlation between RA risk and body mass index (BMI) categories > 25, are in line with previous studies which found that elevated BMI and overweight increase disease risk, with obesity possibly accounting for the rising RA incidence in the last decades [40]. This evidence and our results highlight the importance of addressing dietary habits and physical activity, which in our study and other studies [41] emerged to be effective in reducing RA risk, with weight maintenance being a primary factor in RA prevention.

In the present work, we focused on diet as an environmental RA determinant. Although several studies have shown either deleterious or protective effects exerted by consumption of certain foods, clear evidence resulting from large-scale cohort studies are sometimes contradictory. To overcome this issue, the present work analyzed data from one of the largest prospective studies ever conducted, the UK biobank cohort study. Importantly, we identified for the first time those subjects who developed RA after their enrolment (i.e., RA onset) and their dietary habits, detected through a food frequency questionnaire (FFQ) concerning 29 different groups of foods and alcohol, were compared with those of subjects who did not develop RA from the beginning of the study until the end of the follow-up. 

The results of our analyses showed that consumption of oily fish, cheese, breakfast cereals, alcohol and coffee led to a decreased risk of developing RA, while tea consumption led to an increased risk. 

Moreover, the comparison of food frequency levels of incident and prevalent cases strengthens the possibility of a beneficial role of oily fish, whose consumption with a moderate frequency (once a week) is associated with a more marked reduction of the risk compared to higher or lower intake. Similar conclusions have been reached by a large dose–response meta-analysis [26]. A cohort study not included in this meta-analysis involving 166,013 women from two prospective cohorts (NHS and NHSII) showed that fish intake may attenuate the strong association existing between smoking and RA in subjects younger than 55 years old; however, no protective effects of fish or marine omega-3 fatty acid intake were observed on overall RA risk, i.e., seropositive and seronegative RA [24]. Reasonably, the protective anti-inflammatory effect exerted by omega 3 could be mitigated by the action of other molecules such as the pro-inflammatory metabolite TMAO (its levels results increased in urines after fish consumption in comparison with red meat) [42]. On the other hand, it should be considered that seafood contaminants, such as polychlorinated biphenyls (PCBs) [22], which have already been associated with an increased risk of RA, might reduce the protective effect of omega 3. Thus, we suggest re-evaluating the observed association by distinguishing consumption of raw fish versus cooked one.

Our results regarding meat consumption do not support the presence of a significant association between the different type of meat and RA risk and are in line with a large prospective population cohort study conducted on the Swedish Mammography Cohort [16] and with a 5-year long multi-center Chinese large scale case–control study conducted on 968 RA patients and 1037 matched controls [17]. Since the multivariable HR is increased in proportion with the weekly intake of meat, it would still be interesting to further investigate the effects of cooking methods, given that a positive relationship between fried meat and fish consumptions and urinary levels of TMAO has been highlighted by previous study [21].

The protective role of high cheese consumption on the risk of developing RA that emerged in our study is further reinforced by the aforementioned comparison between incident and prevalent cases. It also supports the results of a prospective cohort study which analyzed data from the Iowa Women’s Health Study, showing that a high intake of vitamin D, both from foods (skim milk, whole milk, ice milk, ice cream, yogurt, cottage cheese, cream cheese, and other cheeses) and supplements, could protect against the risk of developing RA [18]. In view of this, the protective effect exerted by a moderate consumption of fatty fish could be also explained by its high vitamin D content. 

Alcohol consumption has both detrimental and protective health effects, which often involves immune system regulation, but they are strictly dependent on the intake level. High level of alcohol intake is associated with the risk of development of chronic diseases [43]. However, alcohol-mediated adaptive immunity modulation has also been associated with a reduction in both severity and incidence of ADs, such as RA, systemic lupus erythematosus, hyper- and hypothyroidism. Our analysis confirmed the already established protective role of low to moderate alcohol consumption with respect to RA onset [34]. Interestingly, this beneficial role has also been supported by the British Society for Rheumatology guidelines [44]. This effect can be partly explained by its ability to attenuate the inflammatory immune response [45]. Moderate alcohol intake in healthy subjects was shown to significantly reduce the allostimulatory capacity of dendritic cells (DCs) and the activation of T cells [46]. The anti-inflammatory effects of alcohol also depend on the type. Red wine, for example, is rich in polyphenols that can also improve the lipid metabolism and endothelial function, acting as an antioxidant. These effects could also be responsible for RA symptom-relief effects and for the observed reduction of disease activity that has been described by some authors [47]. 

Tea and coffee have been heavily investigated for their possible effect on the development of ADs. With respect to coffee consumption, Karlson et al. did not find any association with either caffeinated or decaffeinated coffee [29], while Heliovaara et al., who evaluated coffee consumption both in a cross-sectional survey and in a cohort study, demonstrated its positive association with the expression of RF [33]. Some years later, a meta-analysis including both mentioned studies concluded that high coffee consumption is associated with an elevated risk of seropositive RA development independently from caffeine intake; however, the association was significant only in the included case–control studies, while in cohort studies only a trend was observed [32]. Subsequently, another cohort study, not included in the meta-analysis, found no association between RA onset and coffee consumption and a minimal association (increased risk) for non-herbal and non-decaffeinated tea consumption [48]. Finally, Rambod et al. reported a protective role for both drinks (particularly green tea and coffee) in their case–control study [49]. Our study suggests that moderate coffee consumption reduces the risk of RA, while high tea consumption increases the likelihood of developing the disease. The opposite effects of these two beverages open the door to a myriad of questions, since they not only both contain caffeine, but caffeine exists in higher quantities in coffee than tea depending on the origin, type and preparation of the drink [50]. Caffeine has anti-inflammatory properties, such as the ability to increase the release of interleukin (IL)-10 (an anti-inflammatory cytokine), reduce neutrophil chemotaxis and inhibit the proliferation of T-helper 1 (TH1) and T-helper 2 (TH2) lymphocytes [51]. Importantly, from our results, a protective effect of caffeine can be speculated, since decaffeinated coffee showed no significant association with the risk of RA; on the other hand, this null association may also be due to a reduction of the statistical power due to a low number of subjects drinking decaffeinated coffee. Beyond caffeine, it is possible that the effects of these beverages on health are also imputable to other substances, such as chlorogenic [52], ferulic and caffeic acids, which have a strong antioxidant power. The latter molecules have been demonstrated to be able to alter DNA methylation and impact the inflammatory cascades [53], while ferulic acid has shown anti-inflammatory effects in rats with collagen induced arthritis [54]. Epigallocatechin-3-gallate (EGCG) is a typical molecule of green tea, which showed immunomodulating abilities by suppressing proliferation of autoreactive T cells, reducing the production of proinflammatory cytokines and inhibiting differentiation of TH1 and TH17 cells [55]. Epicatechin (EC) is also found in green tea, together with EGCG, which accounts alone for 50–80% of the total catechins present in green tea. However, EC has been shown to be able to interfere with the anti-inflammatory cascade promoted by EGCG [56]. Nevertheless, we also recommend distinguishing each type of tea (green, black, etc.) and to re-evaluate the association that appears to be strong also considering the post hoc analysis on prevalent cases. Lastly, the habit of adding sugar to these drinks must be considered, because sugar per se could decrease the protective properties of the substances contained in these two beverages, and act as an inflammation booster [57]. The detrimental role of simple carbohydrates has indeed been stressed in the introduction in the context of sugar-sweetened beverages, which represent one of the few foods, which have been indicated as a hazardous factor in one of the most up to date review on the topic [9]. In our study, we did not evaluate the risk correlated with the consumption of this type of beverages since it was not included in the touchscreen questionnaire administered in the enrolment phase of the UK biobank study. 

We also found that the consumption of breakfast cereals decreases the risk of developing RA. This effect could be ascribable to the presence of fiber which, as already mentioned, contributes to the well-being of the intestinal microbiota [58], and is able to influence systemic inflammation, being negatively related with high-sensitivity C-reactive protein (hs-CRP), IL-6, and tumor necrosis factor-alpha receptor-2 (TNF-alpha-R2) [59]. However, it would be useful to re-evaluate the observed association by differentiating the types of consumed cereals. The protective effects assigned to the high levels of fibers and the low content of simple sugars typical of natural oats, bran or unrefined cereals could in fact be lost if cereals are heavily processed or if they are high in simple sugars and additives, as these conditions could instead exert a proinflammatory action. Thus, the observed association between breakfast cereals and RA onset need to be scrutinized. However, it is noteworthy that our data are, to the best of our knowledge, the first ones linking this food category to the risk of developing RA.

Despite the anti-inflammatory properties attributed to foods rich in fiber, such as fruits, vegetables, legumes, and unrefined cereals, which also contain flavonoids, carotenoids, and other biologically active molecules with anti-inflammatory power widely suggested for the prevention of chronic inflammatory diseases [60], an unexpected positive trend emerged between the consumption of cooked vegetables and the risk of onset of RA. Moreover, it was observed only for cooked and not raw vegetables. An almost opposite result was obtained with the case–control study conducted on 145 RA patients and 188 hospital-based controls, which found an inverse association between both cooked vegetables and olive oil consumption and RA risk, while no associations with raw vegetables were discovered [27]. On the other hand, a cohort study, conducted in 2017 on the NHS I and NHS II, did not find any significant association between RA risk and vegetable consumption [28]. This result is highly unexpected and needs to be evaluated in depth. We speculate that cooking methods (including boiling, grilling and microwaving), may be responsible for the degradation of vitamins, especially vitamin C and B [61]. However, it is unlikely that this degradation, per se, is responsible for an increase in risk. We could rather expect raw vegetables to be more dangerous with respect to the risk of developing RA, since they could (if not washed correctly) contain pesticide residues. In fact, some studies linked the use of pesticides (and glyphosate, still frequently used in agriculture) to an increased risk of RA too [62]. 

The one concerning cooked vegetables certainly remains the most peculiar result, and also in this case the cooking method should be investigated in future studies the 24 h recall in order to acquire more specific data (fried or heavily seasoned vegetables could in fact cause more harm than good). Moreover, the hypothesis of a possible misinterpretation of the term “vegetables” by participants should be considered. Starchy vegetables, such as potatoes, have different nutritional properties in comparison with non-starchy vegetables. Indeed, it has been evaluated that potatoes, especially when not well cooked and peeled, are high in solanine, a glycoalkaloid that has been associated with increased intestinal permeability which results in higher RA risk [6].

Our study has some limitations: first, the FFQ investigated only the main food groups without discriminating between different sub-categories (including for example meat or processed meat) and cooking methods. Secondly, the lack of information on the frequency of consumption of single foods made it impossible to perform the calculation of dietary indexes, which could have helped in adjusting the effect of single food groups on RA risk for the diet; therefore, residual confounding may affect our estimates. Thirdly, despite the demonstrated long-term reproducibility of the FFQ, some study participants could have changed the dietary habits reported at baseline before disease onset, partly affecting our secondary study. Moreover, RA cases were identified through the record linkage with routinely collected clinical healthcare databases that do not allow the evaluation of the reliability of diagnostic criteria used to assess RA. However, a good accuracy of diagnosis of RA based on ICD-10 coding system was demonstrated with positive predictive value of 80% [63]. Lastly, some of our results are in contrast with the body of evidence produced so far in the nutritional immunology field, but it is quintessential to point out that the significant associations emerging in such a large cohort are encouraging and deserve to be deepened and integrated in future studies.

## 5. Conclusions

Dietary habits are considered determinant factors in RA. Our study, based on one of the largest cohorts in which nutritional information is available, highlights the importance of further studies to investigate the role of diet in the development and prevention of RA. Starting from these preliminary data, it will be possible to focus more deeply on the individual foods that have been associate with an increased or decreased risk of RA, in the hope of obtaining solid evidence necessary to draw up new guidelines for RA prevention.

## Figures and Tables

**Figure 1 nutrients-14-01554-f001:**
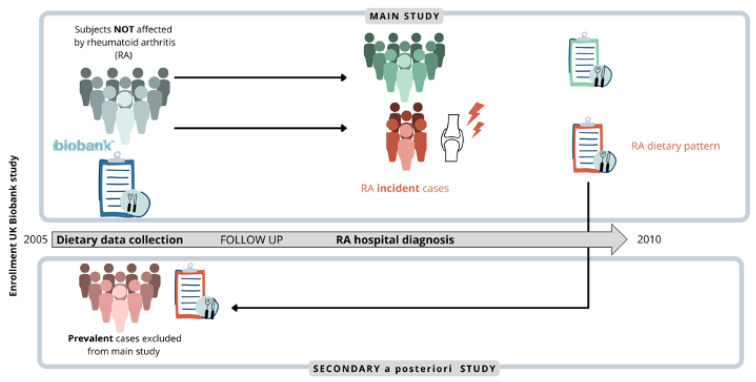
The analysis was aimed at finding an association between certain food items and the risk of rheumatoid arthritis (RA) onset and was conducted considering only RA new diagnosis (incident cases, shown in bold) as outcomes. The observed associations were further explored comparing the distribution of food frequency levels in incident and prevalent cases (people already affected by RA at enrollment, shown in bold).

**Figure 2 nutrients-14-01554-f002:**
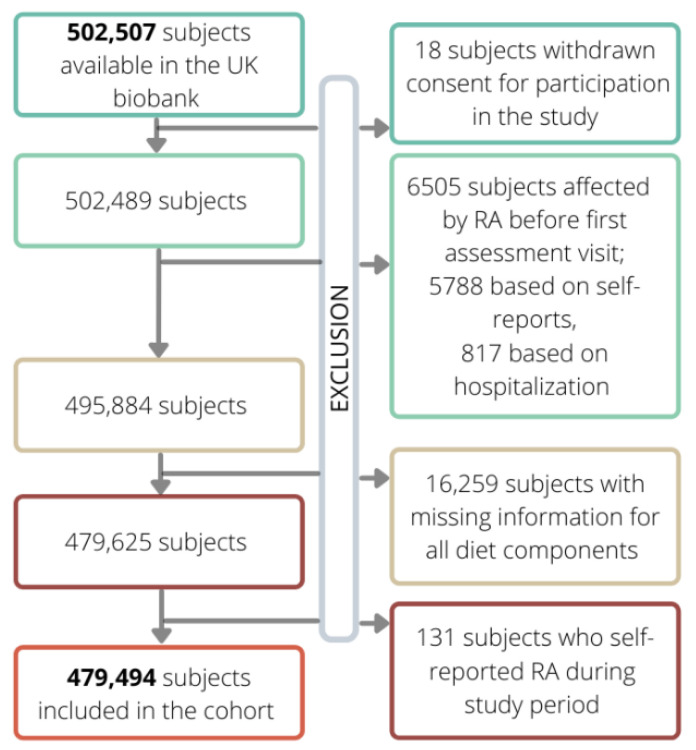
Flow chart of exclusion criteria which led to the definition of the final cohort from the initial one (shown in bold).

**Figure 3 nutrients-14-01554-f003:**
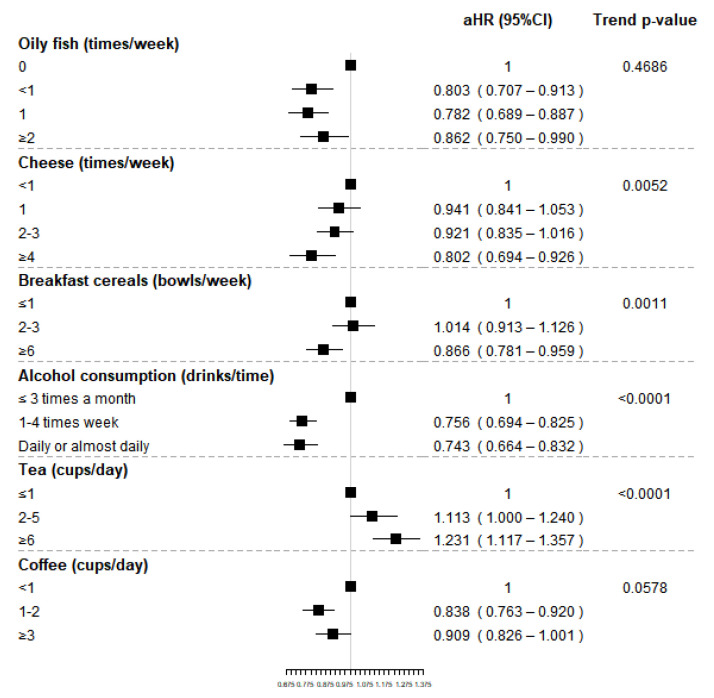
Adjusted hazard ratios (HR) and corresponding 95% confidence intervals (CI) for the association between selected food groups (shown in bold) and RA and *p*-value of the trend test.

**Table 1 nutrients-14-01554-t001:** Distribution of the main characteristics of the cohort members, hazard ratios (HR) and corresponding 95% confidence intervals (CI) for the association with rheumatoid arthritis (RA).

		Rheumatoid Arthritis (RA)		
	Overall	No	Yes	
	*N* = 479,494	*N* = 476,675	*N* = 2819	HR (95%CI)
	*N* (%)	*N* (%)	*N* (%)	
Sex				
Females	259,709 (54.16)	257,827 (54.09)	1882 (66.76)	1
Males	219,785 (45.84)	218,848 (45.91)	937 (33.24)	0.587 (0.543–0.635)
Age				
<50	113,569 (23.69)	113,241 (23.76)	328 (11.64)	1
[50–60)	159,718 (33.31)	158,896 (33.33)	822 (29.16)	1.783 (1.569–2.026)
[60–70)	203,923 (42.53)	202,276 (42.43)	1647 (58.42)	2.837 (2.520–3.194)
≥70	2284 (0.48)	2262 (0.47)	22 (0.78)	3.412 (2.216–5.254)
Area				
England	425,147 (88.67)	422,517 (88.64)	2630 (93.30)	1
Scotland	34,470 (7.19)	34,356 (7.21)	114 (4.04)	0.426 (0.353–0.515)
Wales	19,877 (4.15)	19,802 (4.15)	75 (2.66)	0.518 (0.412–0.653)
Townsend deprivation index				
Q1	119,829 (25.03)	119,246 (25.05)	583 (20.07)	0.623 (0.561–0.691)
Q2	119,649 (24.98)	119,019 (25.00)	630 (22.37)	0.680 (0.615–0.753)
Q3	119,700 (24.99)	119,009 (25.00)	691 (24.54)	0.754 (0.683–0.883)
Q4	119,721 (25.00)	118,809 (24.95)	912 (32.39)	1
Smoking				
Current smokers	37,322 (7.82)	36,993 (7.79)	329 (11.73)	1.779 (1.569–2.016)
Current occasional smokers	13,080 (2.74)	12,990 (2.74)	90 (3.21)	1.404 (1.131–1.743)
Past smokers	110,361 (23.12)	109,534 (23.07)	827 (29.49)	1.530 (1.393–1.679)
Past occasional smokers	124,811 (26.14)	124,194 (26.16)	617 (22.00)	1.012 (0.915–1.121)
Never smokers	191,849 (40.18)	191,022 (40.24)	827 (29.49)	1
BMI categories (kg/m^2^)				
Underweight (<18.5)	2417 (0.51)	2406 (0.51)	11 (0.39)	1.110 (0.612–2.014)
Normal weight (18.5–24.9)	155,221 (32.54)	154,584 (32.60)	637 (22.68)	1
Overweight (25.0–29.9)	203,259 (42.61)	202,140 (42.63)	1119 (40.15)	1.340 (1.216–1.477)
Obese (≥30.0)	116,099 (24.34)	115,079 (24.27)	1020 (36.60)	2.143 (1.941–2.366)
Moderate/vigorous physical activity (days/week)				
0	58,834 (12.54)	58,299 (12.50)	535 (19.87)	1
1–3	174,747 (37.25)	173,947 (37.29)	800 (29.72)	0.507 (0.455–0.566)
4–6	147,347 (31.41)	146,568 (31.42)	779 (28.94)	0.588 (0.527–0.656)
7	88,202 (18.80)	87,624 (18.79)	578 (21.47)	0.732 (0.651–0.823)
Alcohol consumption (drinks/time)				
≤3 times a month	145,185 (30.28)	144,017 (30.21)	1168 (41.43)	1
1–4 times week	235,440 (49.10)	234,281 (49.15)	1159 (41.11)	0.606 (0.558–0.657)
Daily or almost daily	98,289 (20.50)	97,804 (20.52)	485 (17.20)	0.609 (0.548–0.677)
Diabetes				
No	471,755 (98.39)	469,020 (98.39)	2735 (97.02)	1
Yes	7739 (1.61)	7655 (1.61)	84 (2.98)	1.91 (1.534–2.368)
Hypertension				
No	354,937 (74.02)	353,156 (74.09)	1781 (63.18)	1
Yes	124,557 (25.98)	123,519 (25.91)	1038 (36.82)	1.663 (1.540–1.795)
Hypercholesterolemia				
No	456,036 (95.11)	453,382 (95.11)	2654 (94.15)	1
Yes	23,458 (4.89)	23,293 (4.89)	165 (5.85)	1.227 (1.048–1.436)
Other autoimmune diseases				
No	473,960 (98.85)	471,210 (98.85)	2750 (97.55)	1
Yes	5534 (1.15)	5465 (1.15)	69 (2.45)	2.140 (1.686–2.717)

**Table 2 nutrients-14-01554-t002:** Distribution of the diet components, unadjusted and adjusted hazard ratios (HR) and corresponding 95% confidence intervals (CI) for the association with RA and *p*-value of the trend test.

Consumption					
FRUIT and VEGETABLES					
Cooked vegetables (tbs/day)	≤1	2	3	≥4	trend *p*-value
N events/non events	463/86,493	817/156,746	797/128,209	693/98,931	
HR (95%CI)	1	0.979 (0.874–1.097)	1.175 (1.048–1.318)	1.330 (1.182–1.496)	
aHR (95%CI)	1	0.922 (0.819–1.038)	1.018 (0.903–1.148)	1.130 (0.999–1.278)	0.0037
Salad/vegetables (tbs/day)	≤1	2	3	≥4	
N events/non events	451/68,007	710/141,580	629/114,133	976/146,425	
HR (95%CI)	1	0.758 (0.674–0.853)	0.836 (0.741–0.944)	1.016 (0.908–1.135)	
aHR (95%CI)	1	0.810 (0.716–0.916)	0.856 (0.754–0.971)	0.961 (0.854–1.082)	0.8797
Fresh fruit (serving/day)	≤1	2	≥3		
N events/non events	963/169,282	776/1342,56	1062/170,928		
HR (95%CI)	1	1.013 (0.922–1.114)	1.084 (0.993–1.183)		
aHR (95%CI)	1	0.941 (0.853–1.038)	0.918 (0.838–1.007)		0.0729
FISH AND MEAT					
Oily fish (times/week)	0	<1	1	≥2	
N events/non events	374/51,386	856/158,216	1011/179,705	552/84,390	
HR (95%CI)	1	0.748 (0.662–0.844)	0.778 (0.691–0.876)	0.905 (0.793–1.031)	
aHR (95%CI)	1	0.803 (0.707–0.913)	0.782 (0.689–0.887)	0.862 (0.750–0.990)	0.4686
Other fish (times/week)	<1	1	≥2		
N events/non events	937/160,162	1391/236,703	474/77,179		
HR (95%CI)	1	1.001 (0.922–1.088)	1.046 (0.937–1.168)		
aHR (95%CI)	1	1.004 (0.921–1.095)	1.032 (0.921–1.157)		0.5844
Processed meat (times/week)	≤1	2–4	≥5		
N events/non events	1155/187,524	816/139,233	841/148,692		
HR (95%CI)	1	0.950 (0.869–1.039)	0.921 (0.843–1.007)		
aHR (95%CI)	1	0.998 (0.908–1.096)	1.054 (0.958–1.159)		0.3068
Chicken, turkey or other poultry (times/week)	<1	1	≥2		trend *p*-value
N events/non events	121/23,761	324/51,162	2369/400,635		
HR (95%CI)	1	1.228 (0.997–1.514)	1.150 (0.958–1.381)		
aHR (95%CI)	1	1.147 (0.922–1.429)	1.143 (0.943–1.386)		0.2980
Beef (times/week)	0	<1	1	≥2	
N events/non events	327/51,145	1245/216,939	911/151,458	323/54,888	
HR (95%CI)	1	0.894 (0.791–1.010)	0.922 (0.813–1.046)	0.884 (0.758–1.031)	
aHR (95%CI)	1	0.951 (0.836–1.082)	0.999 (0.873–1.144)	1.024 (0.871–1.205)	0.3557
Lamb/mutton (times/week)	0	<1	≥1		
N events/non events	523/82,872	1480/269,629	786/120,816		
HR (95%CI)	1	0.878 (0.794–0.970)	1.038 (0.930–1.160)		
aHR (95%CI)	1	0.908 (0.818–1.008)	0.992 (0.883–1.115)		0.8123
Pork (times/week)	0	<1	≥1		
N events/non events	506/80,295	1483/270,575	803/122,645		
HR (95%CI)	1	0.873 (0.789–0.965)	1.050 (0.940–1.174)		
aHR (95%CI)	1	0.937 (0.842–1.042)	1.038 (0.923–1.168)		0.3083
OTHER FOOD GROUPS					
Cheese (times/week)	<1	1	2–4	≥4	
N events/non events	696/95,256	640/101,778	1188/215,115	287/62,993	
HR (95%CI)	1	0.859 (0.771–0.956)	0.756 (0.689–0.831)	0.628 (0.547–0.720)	
aHR (95%CI)	1	0.941 (0.841–1.053)	0.921 (0.835–1.016)	0.802 (0.694–0.926)	0.0052
Bread (slices/week)	≤1	2	3	≥4	
N events/non events	79/14,845	518/80,479	913/150,723	1253/224,858	
HR (95%CI)	1	1.204 (0.950–1.526)	1.128 (0.897–1.420)	1.035 (0.824–1.299)	
aHR (95%CI)	1	0.998 (0.897–1.110)	0.927 (0.825–1.042)	0.966 (0.857–1.089)	0.3256
Breakfast cereals (bowls/week)	≤1	2–4	≥6		
N events/non events	625/100,393	969/158,200	1214/216,138		
HR (95%CI)	1	0.982 (0.888–1.086)	0.898 (0.815–0.989)		
aHR (95%CI)	1	1.014 (0.913–1.126)	0.866 (0.781–0.959)		0.0011
Consumption					
BEVERAGES					
Alcohol consumption (drinks/time)	≤3 times a month	1–4 times week	Daily or almost daily		trend *p*-value
N events/non events	1168/144,017	1159/234,281	485/97,804		
HR (95%CI)	1	0.606 (0.558–0.657)	0.609 (0.548–0.677)		
aHR (95%CI)	1	0.756 (0.694–0.825)	0.743 (0.664–0.832)		<0.0001
Tea (cups/day)	≤1	2–4	≥4		
N events/non events	650/124,195	785/140,221	1375/211,059		
HR (95%CI)	1	1.074 (0.968–1.191)	1.249 (1.137–1.371)		
aHR (95%CI)	1	1.113 (1.000–1.240)	1.231 (1.117–1.357)		<0.0001
Coffee (cups/day)	<1	1–2	≥3		
N events/non events	923/138,300	1004/185,400	877/151,727		
HR (95%CI)	1	0.816 (0.746–0.892)	0.865 (0.788–0.948)		
aHR (95%CI)	1	0.838 (0.763–0.920)	0.909 (0.826–1.001)		0.0578
Decaffeinated coffee (cups/day)	<1	1–2	≥3		
N events/non events	45/6534	230/35,399	192/29,021		
HR (95%CI)	1	0.954 (0.687–1.301)	0.960 (0.694–1.328)		
aHR (95%CI)	1	0.962 (0.687–1.349)	0.957 (0.679–1.349)		0.8375
Ground coffee (cups/day)	<1	1–2	≥3		
N events/non events	43/9416	205/49,061	95/26,479		
HR (95%CI)	1	0.921 (0.663–1.280)	0.786 (0.548–1.127)		
aHR (95%CI)	1	0.904 (0.647–1.262)	0.792 (0.549–1.143)		0.1770
Instant coffee (cups/day)	<1	1–2	≥3		
N events/non events	104/16,765	534/96,518	566/94,133		
HR (95%CI)	1	0.894 (0.724–1.103)	0.965 (0.783–1.190)		
aHR (95%CI)	1	0.870 (0.701–1.080)	1.000 (0.807–1.239)		0.2082
Other coffee (cups/day)	<1	1–2	≥3		
N events/non events	11/1068	35/4006	20/1656		
HR (95%CI)	1	0.857 (0.435–1.687)	1.169 (0.560–2.440)		
aHR (95%CI)	1	0.720 (0.361–1.437)	0.853 (0.388–1.875)		0.7910

tbs: heaped tablespoons, aHR: Adjusted hazard ratio, adjusted for age, sex, study area, income, smoking, BMI, moderate/vigorous physical activity, comorbidities (type II diabetes, hypertension, hypercholesterolemia and other autoimmune diseases) and alcohol consumption.

**Table 3 nutrients-14-01554-t003:** Comparison of the distribution of food frequency levels of incident and prevalent RA cases.

	Incident RA Cases	Prevalent RA Cases	Disease Free Subjects	*pOR* (95%CI)
	*N =* 2819	*N* = 6604	*N* = 476,675	Prevalent vs. Incident RA Cases
Oily fish (times/week)				
0	374 (13.39)	844 (12.90)	51,386 (10.85)	1
1	856 (30.65)	2079 (31.78)	158,216 (33.40)	1.040 (0.809–1.337)
2	1011 (36.20)	2367 (36.19)	179,705 (37.94)	0.874 (0.681–1.122)
≥3	552 (19.76)	1251 (19.13)	84,390 (17.81)	0.965 (0.734–1.270)
Missing	26	63	2978	
Cheese (times/week)				
≤1	696 (7.76)	1525 (24.16)	95,256 (20.05)	1
2	640 (22.77)	1496 (23.70)	101,778 (21.42)	1.005 (0.800–1.262)
3	1188 (42.26)	2665 (42.23)	215,115 (45.27)	0.944 (0.772–1.153)
≥4	287 (10.21)	625 (9.90)	62,993 (13.26)	0.796 (0.583–1.088)
Missing	8	293	1533	
Breakfast cereals (bowls/week)				
≤1	625 (22.26)	1379 (21.02)	100,393 (21.15)	1
2–5	969 (34.51)	2090 (31.86)	158,200 (33.32)	0.939 (0.756–1.164)
≥6	1214 (43.23)	3090 (47.11)	216,138 (45.53)	1.070 (0.873–1.311)
Missing	11	45	1944	
Tea (cups/day)				
≤1	650 (23.13)	1479 (23.23)	124,195 (26.12)	1
2–3	785 (27.94)	1777 (27.91)	140,221 (29.49)	0.887 (0.712–1.105)
≥4	1375 (48.93)	3110 (48.85)	211,059 (44.39)	0.917 (0.754–1.114)
Missing	9	238	1200	

## Data Availability

The datasets generated/and or analyzed in the current study will be made available for bona fide researchers who apply to use the UK Biobank data set by registering and applying at http://www.ukbiobank.ac.uk/register-apply, accessed date: 5 August 2020.

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
