# Peer review of "Nutrition and Rheumatoid Arthritis Onset: A Prospective Analysis Using the UK Biobank"

_nutrients, 2022, doi:10.3390/nu14081554_

Round 1

Reviewer 1 Report

The authors argued there was limited information and knowledge on the role of diet and food in the onset of rheumatoid arthritis (RA), which is not the case. This study presented data using the UK biobank study to assess food groups in relation to incident RA. They found consumption of coffee, cheese, breakfast cereals, and oily fish was related to a lower risk of incident RA while consumption of cooked vegetables or tea increased the risk. Yet, other food groups, such as different types of meat, processed meat, or fresh vegetable intake, were not related to the diseases. These results are spurious - I think this is due to the lack of managing several methodological issues in the design and analysis of the study. 

  1. Many other factors can affect the diet-diseases relationship. Covariates such as overall dietary quality and comorbidity such as other arthritis, chronic pain, and autoimmune diseases need to be controlled or addressed in the data analyses.
  2. It is unclear how the diet was assessed at baseline and whether the follow-up of dietary assessments was addressed to determine their roles in the onset of RA. The authors did not even mention the length of follow-up time between the dietary evaluation (s) and the incidence of RA.
  3. The authors mentioned that subjects with RA prevalence were removed from the analysis. Yet, they compared the distribution of food frequency between those with the incident and prevalent RA in Table 3. What did these data tell us?
  4. The authors avoided explaining their results, some of which are very different from our knowledge and a large body of evidence in nutritional epidemiology. For example, higher alcohol consumption is generally associated with a higher risk of various chronic diseases, and vegetable intake is generally beneficial to health. What made the results differ in the onset of RA in this study sample?
  5. There are numerous grammatical errors and misspellings throughout the manuscript. 

Author Response

Reply to reviewer 1.

The authors argued there was limited information and knowledge on the role of diet and food in the onset of rheumatoid arthritis (RA), which is not the case. This study presented data using the UK biobank study to assess food groups in relation to incident RA. They found consumption of coffee, cheese, breakfast cereals, and oily fish was related to a lower risk of incident RA while consumption of cooked vegetables or tea increased the risk. Yet, other food groups, such as different types of meat, processed meat, or fresh vegetable intake, were not related to the diseases. These results are spurious - I think this is due to the lack of managing several methodological issues in the design and analysis of the study. 

We thank the reviewer 1 for the suggestions and we would like to emphasize that our aim was not to fix a knowledge gap but rather to overcome the heterogeneity of results and design of the numerous studies available to date on the topic (extensively discussed in the introduction).

  1. Many other factors can affect the diet-diseases relationship. Covariates such as overall dietary quality and comorbidity such as other arthritis, chronic pain, and autoimmune diseases need to be controlled or addressed in the data analyses.

According to reviewer suggestions, in the current version of the manuscript we reported the hazard ratios (HR) derived from the multivariable Cox model further adjusted for type II diabetes, hypercholesterolemia, hypertension and other autoimmune diseases (psoriasis, multiple sclerosis, lupus erythematosus and inflammatory bowel disease). Methods results, and tables were modified accordingly. Unfortunately, the baseline dietary evaluation of enrolled subjects conducted in the UK biobank consisted in a food frequency questionnaire asking about the frequency of intake over the last year of a limited number of food items/food groups, without quantitative information such as portion sizes. Moreover, this general baseline evaluation did not include some items (e.g. legumes/olive oil) necessary to calculate the adherence to the best known pre-determined healthy dietary patterns (e.g. Mediterranean Diet). For this reason, it was not possible to adjust the association estimates for the overall dietary quality. This represents a limitation of our study, which is now acknowledged in the discussion section.

  1. It is unclear how the diet was assessed at baseline and whether the follow-up of dietary assessments was addressed to determine their roles in the onset of RA. The authors did not even mention the length of follow-up time between the dietary evaluation (s) and the incidence of RA.

We thank the reviewer for his/her comments. To better explain how the diet was investigated, we have supplemented the paragraph on dietary data collection with the following information about dietary assessment in UK biobank study protocol.

Starting from 2006 (baseline) until 2010 (end of the enrollment phase) dietary habits of study participants were assessed through the above-mentioned food frequency questionnaire. 

From 2009 to 2010 (enrollment phase) and from 2011 to 2012 (follow up phase), the study protocol was enhanced with the Oxford WebQ, a web-based 24-h dietary assessment tool referring to participant’s dietary intake during the previous 24 h and aimed at evaluating but actual food consumption of up to 206 types of foods and 32 types of drinks. This interview was aimed at estimating the actual dietary habits and was completed at the assessment center by participants during the enrollment phase and remotely after the recruitment period (between February 2011 and June 2012). Since our aim was to investigate the usual long-term diet before RA onset rather than actual dietary habits at enrollment, in the present study we only considered data deriving from the FFQ. Noteworthy, these data were, available for 502,640 subjects. Conversely, the 24h recall information were available only for 210,126 subjects (please, see reference 44, Bradbury, K.E et al.). We thank the reviewers for pointing out the missing information concerning the length of follow up between baseline dietary evaluation and the incidence of RA. We implemented the in the results section of the revised manuscript by reporting the average follow-up time (8.15 year per subject) and the incidence rate of RA (of 7.22 RA events per 10,000 person-year).

  1. The authors mentioned that subjects with RA prevalence were removed from the analysis. Yet, they compared the distribution of food frequency between those with the incident and prevalent RA in Table 3. What did these data tell us?

Our main analysis was intended to prospectively evaluate the role of diet in those subjects who provided nutritional information and were not affected by RA at enrollment but developed the disease later (incident cases). Thus, for the main analysis we excluded subjects already affected by RA (prevalent cases) at enrollment. This was done to avoid confounding factors such as therapy or other. However, since the long-term reproducibility of the main dietary touchscreen variables, as well as the capability of the FFQ to reliably rank participants according to the intake of main foods, was confirmed by researchers of the Nuffield Department of Population Health, University of Oxford [please see ref 44], stating for a stable diet over time, we hypothesized dietary habits related to RA onset, could be found also in the independent cohort of prevalent cases. To test this hypothesis, the available nutritional information from prevalent cases were used, to see if these subjects showed distribution of intake of cheese, breakfast cereals, oily fish, coffee and tea, similar to those who developed RA later. Results confirmed association of oily fish and cheese. To better explain the rationale of our analyses, we have integrated our work with a graphical abstract and modified Figure 1, with the aim of rendering it more intuitive and appealing.

  1. The authors avoided explaining their results, some of which are very different from our knowledge and a large body of evidence in nutritional epidemiology. For example, higher alcohol consumption is generally associated with a higher risk of various chronic diseases, and vegetable intake is generally beneficial to health. What made the results differ in the onset of RA in this study sample?

We apologize if our data were not fully explained in relation with those published. Our results suggest that a moderate, not high, alcohol consumption has a protective effect on RA onset. These protective effect of moderate alcohol consumption with respect to chronic diseases and autoimmune diseases (e.g. RA, systemic lupus erythematosus, hyperthyroidism, and hypothyroidism) risk are in line with a large body of evidences (please see new refs 40,51,52,54,55 and 53) and with the the British Society for Reumatology guidelines. Concerning cooked vegetables, the CI for the observed increased risk was borderline; after adjusting for comorbidities, the association become not significative. We further underlined that the emerged positive trend was unexpected and conflicting with present evidence; we strongly believe that this result deserves to be deepened, as it came out from one of the largest cohorts available to date. In this regard, as a future perspective, we underlined the importance of deepening these results by differentiating between different vegetables types and evaluating the role of the different cooking methods on vegetables’ nutritional proprieties, to permit a correct interpretation

  1. There are numerous grammatical errors and misspellings throughout the manuscript.

A native English speaker has checked the revised manuscript.

Reviewer 2 Report

The relationship of diet to body weight should be discussed within the study results and the discussion. Apart from tables figures depicting the results should also be included.

  1. The article needs extensive English language makeover.
  2. The authors should acknowledge within the manuscript the weak effects of their research project.
  3. It should be stated that the research project is retrospective.
  4. The statistics should be more rigorous.
  5. Extensive discussion acknowledging and comparing with the results of other authors on the same subject should be made within the discussion.  

Author Response

Reply to reviewer 2.

The relationship of diet to body weight should be discussed within the study results and the discussion. Apart from tables figures depicting the results should also be included.

In agreement with reviewer’s suggestions, we discussed the inverse association observed between increasing BMI categories and Rheumatoid Arthritis risk highlighting that our results are in line with previous studies which found that elevated BMI and overweight increase disease risk (please see refs  45-47) and reiterate the importance of managing diet and lifestyle for weight maintenance in RA primary prevention.

We thank the reviewer for the suggestion to include a figure depicting the adjusted hazard ratios (HR) and corresponding 95% confidence intervals (CI) for the association between selected food groups and RA onset (Figure 3)

  1. The article needs extensive English language makeover.

A native English speaker has checked the revised manuscript.

  1. The authors should acknowledge within the manuscript the weak effects of their research project.

We thank the reviewer for his/her suggestion; we added a paragraph discussing limitations of our study. They include the intrinsic low accuracy of the food frequency questionnaire completed by study participants at baseline. The FFQ tool is aimed at investigating subject’s usual diet over the last year, without discriminating between different food sub-categories (including for example meat or processed meat) and cooking methods, which could however be useful to better interpret observed associations. Secondly. the lack of information on the frequency of consumption of single foods made impossible the calculation dietary indexes which could help to adjust the effect of single food groups on RA risk for the diet. Therefore, residual confounding may affect our estimates.  Thirdly, despite the demonstrated long-term reproducibility of the FFQ (please, see reference 44, Bradbury, K.E et al.), some study participants could have changed the dietary habits reported at baseline before disease onset. Finally, some of our results are in contrast with the body evidence produced so far in nutritional immunology field, but the fact remains that significant associations emerging in such a large cohort are encouraging and deserve to be deepened and integrated in future studies.

  1. It should be stated that the research project is retrospective.

We agree with the reviewer ‘s comment that our study is done retrospectively compared to enrollment (i.e based on data collected in the past), but employs data obtained prospectively. It is indeed done on a prospective cohort study based on data collected in the UK Biobank, as we now stated in the Method section. UK Biobank is an open access cohort including half a million middle-aged men and women recruited from the UK between 2006 and 2010 and prospectively followed from 2010 onward via linkage to health and death registries. Participants were enrolled at assessment centers and were asked to complete a touchscreen questionnaire to collect information on sociodemographic characteristics, diet and other lifestyle exposures, general health, and medical history; we exploited the collected data to perform our analysis to study the association between diet and RA risk. To avoid misunderstandings regarding the prospective nature of the UK biobank cohort, we added in the Methods more details regarding the study protocol and the time points in which dietary evaluations were performed. Moreover, we specified that subjects were followed from the date of the first assessment visit, in which nutritional information were collected, until the earliest date among the following events, whatever came first: i) hospitalization with diagnosis of RA, ii) death, iii) migration and iv) end of follow-up (31st March 2017).

  1. The statistics should be more rigorous.

In the revised manuscript, we further adjusted the association’s estimates by type II diabetes, hypercholesterolemia, hypertension and other autoimmune diseases (psoriasis, multiple sclerosis, lupus erythematosus and inflammatory bowel disease). Moreover, we added the average follow-up time (8.15 year per subject) and the incidence rate of RA (of 7.22 RA events per 10,000 person-years). Unfortunately, the lack of information regarding the intake of single foods did not allow the estimation of dietary quality indexes, thus, it was not possible to adjust the association estimates for the overall diet. This represents a limitation of our study which is now acknowledged in the discussion section.

  1. Extensive discussion acknowledging and comparing with the results of other authors on the same subject should be made within the discussion.  

As suggested by the reviewer we further discussed our results. Beside discussing the inverse association observed between increasing BMI categories and RA, we discussed the protective effect of oily fish, already suggested by a large dose-response meta-analysis conducted in 2014.

Based on previous studies, we also underlined the importance to further investigate the role of meat consumption, indicated as hazardous by previous studies, which resulted to increase the multivariable HR for RA in proportion to the weekly consumption, even not significantly. We deemed important to highlight that our results, indicating a protective effect of moderate alcohol consumption are in line with a large body of evidence (please see refs 40,51,52-55) and with the British Society for Rheumatology guidelines (ref.53). After adjusting for comorbidities (type II diabetes, hypertension, hypercholesterolemia and other autoimmune diseases, namely, psoriasis, multiple sclerosis, lupus erythematosus and inflammatory bowel disease), the association between cooked vegetables and RA risk become not significative. However, a positive trend between cooked vegetables consumption and RA risk could still be appreciated. In this regard, further emphasized that the emerged positive trend was unexpected and conflicting with present evidence. We still believe that this result deserves to be deepened, as it came out from a one of the largest cohorts available to date. In this regard, as a future perspective, we underlined the importance of deepening these results by differentiating between different vegetables types and evaluating the role of the different cooking methods on vegetables’ nutritional proprieties, in order to better interpret the observed trend.

Reviewer 3 Report

The authors should provide diagnostic criteria of RA and appropriate references.

Author Response

Dear Prof. Maria Luz Fernandez

Editor-in-Chief

Nutrients

Please find enclosed the revised manuscript (ID: nutrients-1613610) “Nutrition and rheumatoid arthritis onset: a prospective analysis using the UK biobank” by Barbero Mazzucca et al., to be submitted for publication in Nutrients as a reply to the Special IssueDiet, Environmental Factors and Autoimmune Diseases: Can We Find a Path for Preventive Lifestyle?”.

We thank both reviewers for their helpful comments and suggestions. We put all our efforts to reply to all of them. All the revisions within the manuscript were marked up in blue.

We hope that our improved manuscript will be suitable for publication in Nutrients and of interest for the readers.

With my best regards,

Annalisa Chiocchetti

Professor of Immunology

Reply to reviewer 3.

The authors should provide diagnostic criteria of RA and appropriate references.

Reply: in the UK biobank, health outcomes are assessed or by self-report of study participants or through a record linkage with the routinely collected clinical healthcare databases such as hospital admission database; specifically, the Hospital Episode Statistics Admitted Patient Care in England, the Patient Episode Database for Wales Admitted Patient Care in Wales and the General Acute Inpatient and Day Case - Scottish Morbidity Record in Scotland, all are managed by the national health service. Given these premises, unfortunately, it is not possible to provide the diagnostic criteria of RA because they may vary from site to site. However, a good accuracy of diagnosis of RA based on healthcare administrative data based on ICD-10 coding system was demonstrated with positive predictive value of 80% (please, see Ref. n. 63). In the current version of the manuscript, we specify this issue in as limitation of our study in the discussion.

Reviewer 4 Report

Dear colleagues,

Thank you for this interesting and well-written article aiming to determine if certain nutrients increase the risk for RA. By using the publicly available data of the UK Biobank you show that oily fish, alcohol, coffee, and breakfast cereals are protective for RA, whereas tea can increase the risk for RA. I have some comments on how to improve your manuscript:

  • I have some concerns on the statistical analysis used in Table 3: I don’t think the use of Chi square gives any information at all. I suggest using binary logistic regression or another appropriate analysis.
  • Results, page 6, line 267: in the paragraph referring to Table 1, you mention daily alcohol consumption, which is not a variable included in Table 1.
  • Fig 2: “131 subjects who self-reported of AR” should be “131 subjects who self-reported RA”
  • Townsend index should be described in methods
  • Table 1: “Part smokers” should be Past smokers, I guess
  • Can you make Table 2 more reader friendly? It’s a bit difficult to read it
  • Discussion, page 15: “. Importantly, our results suggest that the protective effect can be attributed to caffeine, since decaffeinated coffee showed no significant association with the risk of RA.” I agree this is a possible interpretation and should be mentioned but you need to tone it down as the number of subjects you have in the group of subjects drinking decaf is low and you probably loose some power.
  • The reference list seems very long. Is it possible to cut some of them?

Author Response

Dear Prof. Maria Luz Fernandez

Editor-in-Chief

Nutrients

Please find enclosed the revised manuscript (ID: nutrients-1613610) “Nutrition and rheumatoid arthritis onset: a prospective analysis using the UK biobank” by Barbero Mazzucca et al., to be submitted for publication in Nutrients as a reply to the Special IssueDiet, Environmental Factors and Autoimmune Diseases: Can We Find a Path for Preventive Lifestyle?”.

We thank both reviewers for their helpful comments and suggestions. We put all our efforts to reply to all of them. All the revisions within the manuscript were marked up in blue.

We hope that our improved manuscript will be suitable for publication in Nutrients and of interest for the readers.

With my best regards,

Annalisa Chiocchetti

Professor of Immunology

Reply to reviewer 4.

Dear colleagues,

Thank you for this interesting and well-written article aiming to determine if certain nutrients increase the risk for RA. By using the publicly available data of the UK Biobank you show that oily fish, alcohol, coffee, and breakfast cereals are protective for RA, whereas tea can increase the risk for RA. I have some comments on how to improve your manuscript:

  • I have some concerns on the statistical analysis used in Table 3: I don’t think the use of Chi square gives any information at all. I suggest using binary logistic regression or another appropriate analysis.

Reply: We thank the reviewer for his/her suggestions. Accordingly, we applied logistic regression model to calculate the prevalence odds ratio useful to compare the prevalence of the intake of food items between incident and prevalent RA cases. In the revised manuscript, we modified both methods and results accordingly.

  • Results, page 6, line 267: in the paragraph referring to Table 1, you mention daily alcohol consumption, which is not a variable included in Table 1.

Reply: We apologize for the mistake.  The variable of alcohol consumption has been added in Table 1.

  • Fig 2: “131 subjects who self-reported of AR” should be “131 subjects who self-reported RA”

Reply:  We apologize for this clerical error, which has now been corrected.

  • Townsend index should be described in methods

Reply: We thank the reviewer for his/her suggestion. In the revised the manuscript, we described the Townsend deprivation index in the method’s section (please see lines 232-237 and ref. 38).

  • Table 1: “Part smokers” should be Past smokers, I guess

Reply: We apologize for this clerical error, which was corrected

  • Can you make Table 2 more reader friendly? It’s a bit difficult to read it

Reply: We thank the reviewer for this comment which could make Table 2 more easy in reading. The table was formatted, according to the reviewer’s suggestion

  • Discussion, page 15: “. Importantly, our results suggest that the protective effect can be attributed to caffeine, since decaffeinated coffee showed no significant association with the risk of RA.” I agree this is a possible interpretation and should be mentioned but you need to tone it down as the number of subjects you have in the group of subjects drinking decaf is low and you probably loose some power.

Reply: we thank the reviewer for his/her useful advice. We modified the sentence at page 15, by discussing that the absence of significant associations between decaffeinated coffee and rheumatoid arthritis could be related to a reduced statistical power due to the low number of subjects drinking decaffeinated coffee.

  • The reference list seems very long. Is it possible to cut some of them?

Reply: We thank the reviewer for his/her suggestion. The number of references was reduced in the revised manuscript.